# fNIRS Assessment during Cognitive Tasks in Elderly Patients with Depressive Symptoms

**DOI:** 10.3390/brainsci13071054

**Published:** 2023-07-11

**Authors:** Min-Ju Kang, Su-Yeon Cho, Jong-Kwan Choi, Young-Soon Yang

**Affiliations:** 1Department of Neurology, Veterans Health Service Medical Center, Seoul 05368, Republic of Korea; 2Veterans Medical Research Institute, Seoul 05368, Republic of Korea; 3OBELAB Inc., Seoul 06211, Republic of Korea; 4Department of Neurology, Soonchunhyang University College of Medicine, Cheonan Hospital, Cheonan 31151, Republic of Korea

**Keywords:** depression, geriatric depression scale, functional near-infrared spectroscopy, fNIRS

## Abstract

This study aimed to investigate differences in prefrontal cortex activation between older adults with and without depressive symptoms during cognitive tasks using functional near-infrared spectroscopy (fNIRS). We examined 204 older participants without psychiatric or neurological disorders who completed the Geriatric Depression Scale, digit span, Verbal Fluency Test, and Stroop test. At the same time, prefrontal cortex activation was recorded using fNIRS. During the Stroop test, significantly reduced hemodynamics were observed in the depressive-symptom group. The mean accΔHbO_2_ of all channel averages was 0.14 μM in the control group and −0.75 μM in the depressive-symptom group (*p* = 0.03). The right hemisphere average was 0.13 μM and −0.96 μM, respectively (*p* = 0.02), and the left hemisphere average was 0.14 μM and −0.54 μM, respectively (*p* = 0.12). There was no significant difference in hemodynamic response (mean accΔHbO_2_) between the two groups during the digit span backward and VFT. In conclusion, reduced hemodynamics in the frontal cortex of the depressive-symptom group has been observed. The frontal fNIRS signal and the Stroop task may be used to measure depressive symptoms sensitively in the elderly.

## 1. Introduction

Depression is a serious health problem among the elderly, with 10% of the elderly presenting significant depression in primary care [1,2]. Depression is a common problem in the geriatric population; however, it is often undetected or undertreated [2]. Atypical symptom presentation and overlap with chronic medical illness make diagnosing depression more challenging [3,4,5]. Furthermore, most elderly with clinically significant depressive symptoms do not meet standard diagnostic criteria for major depression disorder (MDD) [6]. These patients are at high risk for subsequent development of major depression, suicidal ideation, and functional impairment [6,7]. Despite limited evidence for treating minor depression, several studies found that pharmacological therapies have modest effects compared with usual care or placebo [8,9]. The availability of clinically useful and cost-effective biomarkers for early detection and differential diagnosis of depression in older adults is essential for patient management.

The present study’s primary objective was to investigate the presence of depressive symptoms, which, while not necessarily enough to fulfill MDD criteria, may impact prefrontal hemodynamic response. Previous studies have sought to compare hemodynamic changes between healthy subjects and MDD. However, MDD participants in most studies were based on operational diagnostic criteria such as DSM-IV or ICD-10 and may not include patients with minor depression or atypical symptoms [10]. Near-infrared spectroscopy (NIRS) has recently been widely studied as a diagnostic technique for MDD [11]. NIRS is a non-invasive optical neuroimaging tool that measures the concentration of oxygenated hemoglobin (oxy-Hb) and deoxygenated hemoglobin in the capillaries and monitors alterations in hemoglobin concentration which reflect changes in the regional cortical blood flow [12]. NIRS allows for continuous monitoring of functional activation during the execution of various cognitive tasks, and it is more robust against motion-related artifacts in a non-invasive method, which is especially beneficial for specific populations that may be difficult to assess, such as infants and the elderly [13,14]. Numerous studies have demonstrated that oxy-Hb does not increase as much in the frontotemporal regions during cognitive tasks in patients with MDD compared to healthy controls [15]. This study screened many healthy older patients who have not been diagnosed with a psychiatric disorder using the Geriatric Depression Scale short form (GDS) [16] to evaluate the presence of depressive symptoms. Furthermore, we performed a comparative analysis of the activation level in the frontal cortex among the normal older adult and depressive symptom groups. We investigated hemodynamic response during various cognitive tasks.

## 2. Materials and Methods

### 2.1. Participants

Two hundred and six people over 60 years old were recruited in the Biobank of the Veterans Medical Research Institute. All participants were recruited amongst volunteers for blood donations and could answer a self-questionnaire and perform cognitive tasks. All participants were interviewed to exclude comorbid medical conditions, neurodegenerative diseases, and psychiatric diseases. Study participants were excluded if they reported cognitive impairment, hearing impairment, severe cardiopulmonary diseases, unstable medical illnesses, or a history of psychiatric disorders. Participants taking antidepressants, antipsychotics, and hypnotics were also excluded. Baseline demographic characteristics and depression symptoms were recorded and assessed by the main researchers during a face-to-face interview.

### 2.2. Assessment of Depression Symptoms

The current study used the Korean version of GDS to assess depressive symptoms [17]. Each of the 15 items was coded as 0 (no) or 1 (yes). The GDS is a self-questionnaire; however, the research nurses read it aloud without additional comments for illiterate participants. The GDS is the most widely used self-rating scale for screening for depression in older patients. It comprises 15 dichotomous depression items with total scores ranging from 0 to 15; higher scores indicate more severe depressive symptoms [18]. In other studies of the GDS, sensitivity and specificity ranged from 79% to 100% and 67% to 80%, respectively, in older adults receiving primary care [19]. The Korean version of GDS was validated for use in Korean elderly, and a score of 8 was suggested as the optimal cutoff point to screen for MDD [17]. Since the GDS scores above 5 best predict depression with poor perceived quality of life, we defined individuals having scores above 5 as having depressive symptoms [20]. All participants provided written informed consent. The study protocol was approved by the Institutional Review Board of Veterans Healthcare Medical Center (2019-03-032).

### 2.3. Experimental Design

According to the result of the GDS, we divided the patients into two groups: control (*n* = 117) and depressed (*n* = 67). Each group performed the digit span backward, Stroop test, and semantic VFT. We recorded the hemodynamic response during the performance of the protocol using a commercial wireless continuous-wave near-infrared spectroscopy system (NIRSIT; OBELAB Inc., Seoul, Republic of Korea) [21]. The digit span backward was used to measure attention and working memory [22]. The task began with a length of 2 in backward, and the participant had to listen and repeat the span backward. A single trial was presented at each length. The test was halted when the participant failed each trial at an equal digit length.

The correctly repeated number of digits was measured. Each task was performed twice, with a 30 s break between tasks. The Stroop test measured mental control and response flexibility [23]. This task measures new responses elicited while suppressing the dominant response. The task involved reading the word of a written letter (word reading) and the color of a letter written in red, blue, yellow, or black (color reading) within a limited time frame (30 s). The number of correctly answered stimuli was measured. VFT involves generating as many words as possible within a certain time frame in a specific category [24]. In this study, participants were instructed to generate as many Korean words as possible that are the name of an animal within 30 s. One type of task was conducted per session, and a 1 min break was provided between each session.

### 2.4. Data Processing

The hemodynamic response of the prefrontal cortex was recorded using a high-density NIRS device (NIRSIT; OBELAB Inc.), which was composed of 24 sources (laser diodes) emitting two wavelengths (780/850 nm) and 32 photodetectors at a sampling rate of 8.138 Hz. The channel separation between the source and the detector was 3, and the total number of channels was 48; the source-detector array is shown in Figure 1a,b. The detected light signals were filtered by a bandpass filter. The cutoff frequency of the low pass filter is 0.1 Hz to minimize physiological noise such as heart-rate and Mayer wave and 0.005 Hz for the high pass filter. During the resting period, low-quality channels with a signal-to-noise ratio of less than 30 dB were rejected prior to hemodynamic data extraction to prevent misinterpretation.

Relative hemodynamic changes in each channel during every trial of the tasks were calculated separately using the modified Beer–Lambert law [25]. The differential pathlength factor (DPF) of 780 nm and 850 nm were 6.0 and 5.2, respectively [26]. The baseline of each trial was defined the averaged value during the 2 sec before the trial begins. The results from multiple trials were block-averaged individually before grand-averaging for each group. The accumulated oxygenated hemoglobin (oxy-Hb/HbO2) values (accΔHbO_2_) during the task period represented the activation of the prefrontal cortex.

### 2.5. Statistical Analysis

The representative means and standard deviations of accΔHbO_2_ were calculated from all right and the left prefrontal cortex. Within-group difference between accΔHbO_2_ of the right and left channels was evaluated using Wilcoxon signed-rank tests since both groups were not normally distributed after evaluating the normality of the data by the Shapiro–Wilk test. Kruskal–Wallis tests evaluated between-group differences of accΔHbO_2_. All statistical analyses were performed using SPSS version 22 (SPSS Inc., Chicago, IL, USA). Statistical significance was set at *p* < 0.05 (two-tailed).

## 3. Results

Among 206 participants, 117 were in control, and 67 were in the depressive symptom group. The average age for the control group was 72 years old, and the depressive-symptom group average was 71 years old, with no significant differences (*p* > 0.05) between groups. There were no statistically significant differences in sex and education between the depressive symptom and control groups. Expectedly, the depressive symptom group had a higher score on the GDS (*p* < 0.001) (Table 1). Comparing group performance on cognitive tasks, the depressive-symptom group had significantly lower performance in digit span backward (*p* > 0.05). Both groups had no significant differences in the VFT and Stroop test (Table 1).

The hemodynamic response during different cognitive tasks is illustrated in Figure 2. During the Stroop test, changes in oxy-Hb concentration during the activation period were shown, as illustrated in Figure 3. Mean accΔHbO_2_ in the right hemisphere, left hemisphere, and both hemispheres was significantly lower in the depressive-symptom group. During the VFT and digit span test, there was no significant difference in both frontal areas between the control and depressive symptom group. Mean accΔHbO_2_ values during the cognitive tasks in each group are shown in Table 2.

## 4. Discussion

This study aimed to investigate the differences in the hemodynamic response of frontal cortices between healthy control and depressive symptom participants in the elderly. The result revealed that participants with depressive symptoms showed a reduced hemodynamic response in both frontal cortices during the Stroop test. Although there was no significant difference in hemodynamic response between the control and depressive symptom group during the digit span backward and VFT, lower mean accΔHbO_2_ in the depressive symptom group is evident, as illustrated in the activation map.

Major depression in the elderly is often accompanied by cognitive impairment spanning multiple domains [27]. Studies have focused on the role of executive functions, such as planning, organizing, initiating, perseveration, sequencing, and attention set-shifting impairment, in geriatric depression. Patients with executive dysfunction are at risk for poor, slow, and unstable antidepressant treatment response and relapse of depression [28,29]. The frontal cortex controls executive functions, emotional regulation, decision-making, and attention control [30]. Reduced activation and connectivity of the dorsolateral prefrontal cortex in depression has been consistently revealed in some studies using various neuroimaging techniques, including functional magnetic resonance imaging (fMRI) and positron emission tomography (PET). Okada et al. revealed decreased left prefrontal activation and reduced task performance in depressed patients utilizing the VFT [31]. A PET study revealed decreased glucose utilization in the prefrontal cortex and limbic/amygdala regions [32].

To date, many studies on the diagnostic applications of NIRS and depression have been conducted. A recent meta-analysis of evidence from 64 studies that used NIRS signals to diagnose MDD revealed a demonstration of attenuated cerebral hemodynamic changes in depressed compared with healthy individuals [33]. In papers that used NIRS to distinguish depressed patients from healthy controls, most studies showed that depressed patients were associated with decreased oxy-Hb in prefrontal regions during cognitive stimulation [33,34]. Our study confirmed a relationship between depressive symptoms and the average accΔHbO_2_ across the frontal lobe. Nonetheless, considering a large number of participants, this finding has the potential to have high reproducibility and to be applied as a sensitive measurement of cognitive control in geriatric depression.

Interestingly, VFT and digit span task showed no statistical significance, while the activation map showed a visible difference between the two groups. Notably, most studies on fNIRS and MDD adopted the VFT as a cognitive stimulus, confirming a reduced frontal fNIRS oxy-Hb response in depression. The digit span task has also been extensively used to assess working memory and attention in neuropsychological research [35], and a previous study using NIRS reported that oxy-Hb was significantly higher during the digit span task than during baseline [36]. Yet, the digit span task has not been widely applied as an activation task in the study of depression. Altogether, this study indicates that VFT and digit span task may not distinguish differences in frontal activation in patients with subsyndromal depression. The color-word Stroop task has been used to study conflict resolution mechanisms and frontal function in many neurological and psychiatric disorders [37,38]. Conflict resolution is one significant role of executive function and is critical in keeping adequate performance with significant interference between multiple sources of information. In the present study, the prefrontal cortex displayed lower activation levels during the color task in the depressive-symptom group than the control group, indicating that the prefrontal cortex could not be recruited in conflict resolution. These findings agree with previous fMRI studies that report prefrontal cortex utilization to cope with Stroop-related interference [39]. In another fMRI study by Simeonova et al., the Stroop test combined with an n-back component in fMRI in MDD patients revealed hypoactivation in lingual gyrus during word reading while hyperactivation of the same structure in color reading, which indicates fungiform gyrus as a structural marker for MDD [40]. Yoon et al. evaluated hemodynamic response using fNIRS with the Stroop task in mild cognitive impairment (MCI). They found a higher hemodynamic response in the right prefrontal cortex of the non-amnestic MCI group, which may be a compensatory effect [41]. As such, evidence supports the utilization of the Stroop test and fNIRS for evaluating cognitive control and neural compensation, or vice versa, during the activation task to differentiate between pseudodementia associated with depression [42] and dementia due to neurodegenerative disease; however, further study is needed. The present study has three notable limitations that should be addressed in future research. First, a global cognitive measure was not conducted for participants. Patients with subjective cognitive and functional impairment were excluded; however, participants’ cognitive status before data collection was not confirmed. This may have resulted in the inclusion of participants with cognitive deficits. Considering the high comorbidity between cognitive disorders and depressive symptoms, our findings should be interpreted cautiously, and future studies should aim to address this limitation by including comprehensive cognitive assessments. Second, neuroimaging assessment did not exclude structural abnormality or neurodegeneration, so we cannot confidently rule out the possible inclusion of participants with depression associated with an organic condition. Third, this study did not control confounding variables such as autonomic dysfunction, diet, physical activity, and medications. Fourth, this study did not use standard questionnaires such as the geriatric depression scale to assess depression. Additional trials with a multimodal assessment of depressive symptoms by face-to-face interviews with specialists, neuroimaging assessment, and neuropsychological tests should be conducted.

## 5. Conclusions

In conclusion, this study provides evidence for reduced prefrontal activation during cognitive control in the elderly with depressive symptoms. The Stroop–fNIRS paradigm shows promise as an objective indicator of depressive symptoms and may aid in diagnosis, assessment, and treatment monitoring in geriatric depression. Future research should control possible variables with larger sample sizes.

## Figures and Tables

**Figure 1 brainsci-13-01054-f001:**
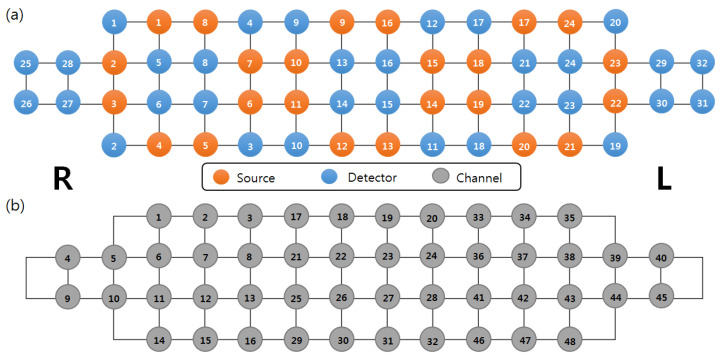
(**a**) Arrangement of sources, detectors, and (**b**) channels.

**Figure 2 brainsci-13-01054-f002:**
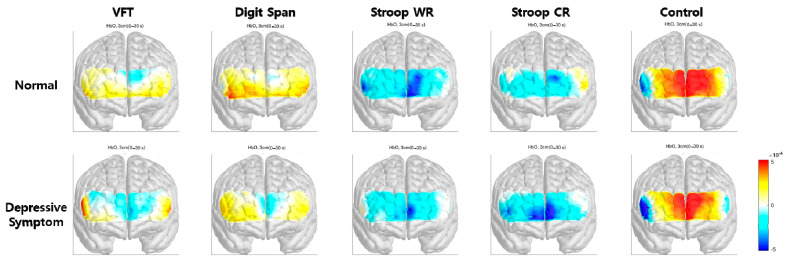
Activation map during the cognitive tasks in normal and depressive symptom group. VFT: verbal frontal task.

**Figure 3 brainsci-13-01054-f003:**
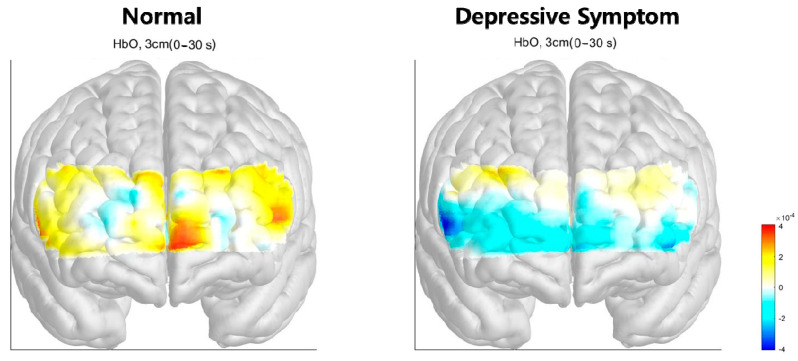
Activation map during the Stroop test (color reading versus word reading) in two groups showing hypoactivation of both frontal cortices in the depressive-symptom group compared to the normal group.

**Table 1 brainsci-13-01054-t001:** Demographics and task performance of cognitive tasks.

	Control	Depressive Symptom	*p*
Number of participants	117	67	
Female, *n* (%)	82 (70.1)	54 (80.6)	0.119
Age, year, median (IQR)	72 (67–74)	71 (67–73)	0.484
Education, year, mean	9.90 (3.65)	8.16 (4.26)	0.203
GDS, mean	1.65 (1.26)	7.47 (2.67)	<0.001
Task performance			
Digit span backward, median (IQR)	4 (3.5–4.0)	4 (3.0–4.0)	0.025
VFT, mean	11.01 (3.31)	10.08 (2.69)	0.061
Stroop WR, median (IQR)	24.5 (20.0–25.0)	23.0 (19.0–25.0)	0.058
Stroop CR, median (IQR)	10.0 (7.0–19.0)	12.0 (8.0–18.0)	0.403
Stroop WR-CR, median (IQR)	10.0 (4.0–16.0)	8.0 (3.0–13.0)	0.089

Data are presented as number (SD) unless otherwise specified. CR, color reading; IQR, interquartile range; SD, standard deviation; VFT, verbal fluency task; WR, word reading may have a footer.

**Table 2 brainsci-13-01054-t002:** Mean accumulated oxygenated hemoglobin (accΔHbO_2_) value during the cognitive tasks.

Cognitive Task	Channel	accΔHbO_2_		t	*p*
		Control	Depressive Symptom		
VFT	All	0.120 (0.560)	0.001 (0.434)	1.608	0.110
	Right hem.	0.146 (0.588)	0.031 (0.457)	1.476	0.142
	Left hem.	0.094 (0.609)	−0.029 (0.503)	1.403	0.162
Digit span	All	0.187 (0.522)	0.098 (0.373)	1.343	0.181
	Right hem.	0.191 (0.559)	0.151 (0.446)	0.505	0.614
	Left hem.	0.184 (0.542)	0.045 (0.393)	1.827	0.069
Stroop WR	All	−0.240 (0.538)	−0.170 (0.456)	−0.964	0.336
	Right hem.	−0.251 (0.613)	0.613 (0.551)	−1.047	0.297
	Left hem.	−0.232 (0.57)	0.570 (0.51)	−0.665	0.507
Stroop CR	All	−0.110 (0.565)	−0.234 (0.459)	1.518	0.131
	Right hem.	−0.122 (0.578)	−0.252 (0.526)	1.438	0.152
	Left hem.	−0.094 (0.631)	−0.221 (0.505)	1.360	0.175
Stroop CR-WR	All	0.134 (0.679)	−0.064 (0.468)	2.308	0.022
	Right hem.	0.130 (0.75)	−0.088 (0.554)	2.205	0.029
	Left hem.	0.138 (0.714)	−0.040 (0.533)	2.000	0.047

Data are presented as number (SD). accΔHbO_2_, Mean accumulated oxygenated hemoglobin; CR, color reading; SD, standard deviation; VFT, verbal fluency task; WR, word reading.

## Data Availability

The data presented in this study are available on request from the corresponding author.

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
