# Peer review of "fNIRS Assessment during Cognitive Tasks in Elderly Patients with Depressive Symptoms"

_brainsci, 2023, doi:10.3390/brainsci13071054_

Round 1

Reviewer 1 Report

Comments and Suggestions for Authors

The review of the manuscript entitled: “fNIRS Assessment During Cognitive Tasks in Elderly Patients with Depressive Symptoms

Comments for Authors:

Thank you for the valuable research you have done. The study concerns interesting topic and acceptable writing. However, there are some issues:

1) ‘Materials and Methods’ section, ‘Assessment of depression symptoms’ subsection: What language version of the GDS scale applied for this study? If translated version has been used, what is the statistical properties of this version?

2) ‘Materials and Methods’ section, ‘Assessment of depression symptoms’ subsection, lines 83 and 84, the authors mentioned: “All participants or their legal representatives provided written informed consent”. In which condition authors decided to obtain informed consent from 'legal representative' instead of the participant him/herself? What was the exact number of participants for whom 'legal representative' signed informed consent?

3) ‘Results’ section: It is recommended that authors present mean value of GDS scores for each group of depressed and control in this section.

4) One important limitation of this study is that authors did not assess the cognitive state of the participants to rule out 'neurocognitive disorders'.  Unfortunately, self-report of denying cognitive impairment from participants is not enough for this purpose. As neurocognitive disorders are highly comorbid with depressive symptoms and even MDD; therefore, the relationship between depression and cognitive test results for addressing primary objective of the study (line 59) and conclusions without omitting or adjusting this confounding factor, would be unclear and unprovable. Unfortunately, this limitation is much bigger than can be resolved by mentioning it in limitations paragraph of the manuscript.

Good luck

Author Response

Response to Reviewers’ Comments

July 24th, 2023

Dear Reviewers,

We greatly appreciate your thoughtful comments, which helped us improve our manuscript's quality and readability. The authors agreed with the points made and have revised the manuscript as requested.

Reviewer 1

1) ‘Materials and Methods’ section, ‘Assessment of depression symptoms’ subsection: What language version of the GDS scale applied for this study? If translated version has been used, what is the statistical properties of this version?

We used the Korean version of the Geriatric Depression Scale short form. Reference to the Korean form of GDS short form was added in the ‘Materials and Methods’ section (Cho et al., Journal of psychosomatic Research, vol 57, issue 3, Sep 2004, Page 297-305, https://doi.org/10.1016/j.jpsychores.2004.01.004). The paper suggested that the cut-off score best predicting major depression disorder was a score of 8. In this presenting study, we targeted patients with depressive symptoms, not MDD patients, and the object of this study is early diagnosis of depression in elderly. Therefore, we adopted a cut-off score of 5 which best predicts depression with a poorly perceived quality of life.

2) ‘Materials and Methods’ section, ‘Assessment of depression symptoms’ subsection, lines 83 and 84, the authors mentioned: “All participants or their legal representatives provided written informed consent”. In which condition authors decided to obtain informed consent from 'legal representative' instead of the participant him/herself? What was the exact number of participants for whom 'legal representative' signed informed consent?

All participants were able to provide written informed consent by themselves. All of them had no limitation in self-reporting their symptoms and answering their questionnaire; therefore, they could sign written informed consent. I edited the line, as shown below.

“All participants provided written informed consent.”

3) ‘Results’ section: It is recommended that authors present mean value of GDS scores for each group of depressed and control in this section.

Thank you for your valuable comment. We added the mean value of GDS scores for each group in Table 1.

4) One important limitation of this study is that authors did not assess the cognitive state of the participants to rule out 'neurocognitive disorders'.  Unfortunately, self-report of denying cognitive impairment from participants is not enough for this purpose. As neurocognitive disorders are highly comorbid with depressive symptoms and even MDD; therefore, the relationship between depression and cognitive test results for addressing primary objective of the study (line 59) and conclusions without omitting or adjusting this confounding factor, would be unclear and unprovable. Unfortunately, this limitation is much bigger than can be resolved by mentioning it in limitations paragraph of the manuscript.

Thank you for your valuable comment. We agree that not assessing the participants’ cognitive state is a limitation in our research. We acknowledge that self-report measures alone are insufficient to rule out neurocognitive disorders definitively. In retrospect, incorporating a comprehensive cognitive assessment would have been valuable in ensuring a more accurate evaluation of cognitive functioning among the participants. In this study's clinical setting, it was impossible to provide the human and temporal environment to perform the cognitive test on the subjects directly.

We have revised the lines in the conclusion to emphasize this limitation more explicitly and acknowledge its impact on interpreting our results.

“Considering high comorbidity between cognitive disorders and depressive symptoms, our findings should be interpreted cautiously, and future studies should address this limitation by including comprehensive cognitive assessments.”

In summary, we revised our manuscript to address the various issues raised by the reviewers. Upon reviewing our revisions, we hope you will find the manuscript acceptable for publication in Brain Sciences.

Sincerely yours,

YoungSoon Yang, MD

May 24th, 2023

YoungSoon Yang, MD

Department of Neurology, Soonchunhyang University College of Medicine, Cheonan Hospital

31 Soonchunhyang 6-gil, Dongnam-gu, Cheonan 31151, Republic of Korea

Tel: +82-41-570-3639

Fax: +82-41-570-3639

Email: astro76@naver.com

Reviewer 2 Report

Comments and Suggestions for Authors

This is an exciting piece of research, which delivers evidence about alteration in brain oxidative metabolism (essentialy this represents hemodynamic response) during performance of various cognitive tasks in functional NIRS session. The authors have investigated depression in geriatric population by means of Stroop Color and Word Test, Verbal Flulency Test and digit span backwards test.

Interestingly the most prominent result indicates that the Stroop test can detect clear difference across the groups. In that context authors should also discuss other contributions using Stroop test (SCWT) in depression with functional MRI paradigms (https://doi.org/10.31083/j.jin2104113).

In my view authors have selected semantic fluency in order to avoid bias with SCWT. However they are advised to better justify for the reader why they have decided to employ semantic instead of phonemic fluency test.

What is really significant about this type of study is its abillity to translate from neuroimaging findings into clinical rationale for diagnosis and prognosis. 

Author Response

Response to Reviewers’ Comments

July 24th, 2023

Dear Reviewer,

We greatly appreciate your thoughtful comments, which helped us improve our manuscript's quality and readability. The authors agreed with the points made and have revised the manuscript as requested.

This is an exciting piece of research, which delivers evidence about alteration in brain oxidative metabolism (essentially this represents hemodynamic response) during performance of various cognitive tasks in functional NIRS session. The authors have investigated depression in geriatric population by means of Stroop Color and Word Test, Verbal Flulency Test and digit span backwards test.

Interestingly the most prominent result indicates that the Stroop test can detect clear difference across the groups. In that context authors should also discuss other contributions using Stroop test (SCWT) in depression with functional MRI paradigms (https://doi.org/10.31083/j.jin2104113).

In my view authors have selected semantic fluency in order to avoid bias with SCWT. However they are advised to better justify for the reader why they have decided to employ semantic instead of phonemic fluency test.

What is really significant about this type of study is its abillity to translate from neuroimaging findings into clinical rationale for diagnosis and prognosis.

Thank you for your valuable opinions. I have added discussion regarding the difference in activation area during stroop color vs. word reading revealed in the fMRI study. This study revealed altered activation during the stroop test in the lingual and fusiform gyri in MDD patients. This study would support our findings of altered activation in the prefrontal area observed in fNIRS.

We appreciate the reviewer's comment and the opportunity to clarify further our choice of employing semantic fluency. The decision to utilize semantic fluency was based on several considerations.

In the present study, we aimed to minimize potential confounding factors arising from our study population’s phonological or executive function deficits. Phonemic fluency tests heavily rely on phonological processing skills and executive functions, such as working memory and cognitive flexibility. By choosing semantic fluency, we aimed to reduce the influence of these factors and focus primarily on the semantic aspects of verbal fluency.

Moreover, previous studies have demonstrated that semantic and phonemic fluency tasks engage distinct cognitive processes and neural networks. Semantic fluency tasks tend to activate anterior temporal regions associated with semantic processing, while phonemic fluency tasks engage prefrontal regions associated with executive functions. By selecting semantic fluency, we intended to examine the specific cognitive mechanisms associated with semantic memory and retrieval.

However, we acknowledge that providing explicit justification for our choice of semantic fluency compared with phonemic fluency would enhance the transparency of our methodology. Moreover, in the clinical setting for this study, we could not provide the human and temporal environment to perform both fluency tests directly.

In summary, we revised our manuscript to address the various issues raised by the reviewers. Upon reviewing our revisions, we hope you will find the manuscript acceptable for publication in Brain Sciences.

Sincerely yours,

YoungSoon Yang, MD

May 24th, 2023

YoungSoon Yang, MD

Department of Neurology, Soonchunhyang University College of Medicine, Cheonan Hospital

31 Soonchunhyang 6-gil, Dongnam-gu, Cheonan 31151, Republic of Korea

Tel: +82-41-570-3639

Fax: +82-41-570-3639

Email: astro76@naver.com

Reviewer 3 Report

Comments and Suggestions for Authors

Thank you for the opportunity to review this work.

However, the manuscript still has the following problems worthy of attention, through the improvement of these problems can better improve the quality of the manuscript.

Abstract

The aim needs to be more specifically stated, e.g. "To investigate differences in prefrontal cortex activation between elderly with and without depressive symptoms during cognitive tasks using functional near-infrared spectroscopy (fNIRS)."

The methods section should concisely state the sample size, inclusion/exclusion criteria, assessment tools/measures and experimental tasks used. For example,  "204 elderly participants without psychiatric or neurological disorders completed the Geriatric Depression Scale, digit span, Verbal Fluency Test and Stroop test while prefrontal cortex activation was recorded using fNIRS."

The conclusion needs to be a strong, compelling summary statement regarding the key implications or significance of the results.

Avoid excessive use of abbreviations in the abstract. Only use if unavoidable and spell out in full at first mention.

Introduction

The first paragraph briefly introduces the background of depression in older adults and challenges in diagnosis and treatment. However, it lacks a clear thesis statement to specify the objectives and focuses of the current study. It is recommended to add a thesis statement at the end of the first paragraph to guide the readers.

The second paragraph should be reorganized. Currently, it introduces NIRS and findings of previous studies first and then states the objectives of the current study. For clarity and logic, it is better to first state the objectives and research questions, and then explain how NIRS can be used to address the research questions with support from previous findings.

The language needs to be further polished with better coherence and flow. Some sentences are too long and difficult to follow. Transition words can be used to enhance the coherence between ideas.

For conciseness, the background information on PET and fMRI in the second paragraph could be removed. The authors could just focus on introducing NIRS and its applications in investigating MDD.

Methods

The method section needs to be substantially expanded to provide sufficient details for replication. Important information such as specific inclusion/exclusion criteria, equipment specifications, and data analysis procedures are missing or not elaborated enough.

For the participants, more demographic information such as age, gender, education level, and socioeconomic status needs to be provided. The sample size for each group should be clearly stated. The inclusion/exclusion criteria should be specified in a list format.

For the GDS, the validity, reliability and scoring procedures of the scale should be briefly described. The cut-off score of 5 for defining depression also needs to be justified with evidence from the literature.

For the experimental design, the specific cognitive tasks, duration, trial numbers and intervals need to be elaborated in detail. The reasons for choosing the specific tasks need to be explained. Diagrams or pictures can be used to illustrate the tasks design and procedures.

For the NIRS system, the full name of the model, spatial resolution, wavelengths used and sampling rate should be provided in the first mention. Additional technical specifications are needed for the sources, detectors and the modified Beer-Lambert law used for data analysis.

For data analysis, the specific statistical tests used for which types of comparisons need to be clearly stated. The reasons for choosing non-parametric tests should be explained based on the Shapiro-Wilk test results. Multiple comparison correction methods should be considered if there are many group comparisons.

Results

The results need to be presented in a logical order in separate paragraphs with smooth transition. It is suggested to start with participant demographics, then cognitive performance and followed by NIRS findings. Currently, the information is mixed together, reducing readability.

For demographics, report mean (standard deviation) for age, and percentage for gender and education. Provide p values to compare groups.

For cognitive performance, report mean (SD, range) scores for each task. Provide test statistics and p values for between-group comparisons.

Check figures for titles.

Discussion and conclusion

The discussion needs to have a clear and logical flow of ideas. Currently, the points are not well-connected, making the discussion appear disorganized and lacking in coherence. It is suggested to start with restating the main findings, then compare with previous literature, discuss implications, limitations, and future directions in separate paragraphs.

For the main findings, restate the differences in prefrontal activation during the Stroop test and the lack of differences during VFT and digit span between groups. Elaborate on the implications of these results in light of what is known about these tasks and prefrontal cortex function. Discuss the potential underlying neural mechanisms. 

Thoroughly compare the current results with findings from previous fNIRS, fMRI and neuropsychological studies on depression, MCI and executive function. Explain how the results support or differ from the existing literature. Discuss the clinical implications and utilities of the Stroop test and fNIRS.

In limitations. Discuss how the lack of neuropsychological assessment, medical evaluation and control of confounding factors may have affected the results and interpretation. Suggest improvements for future research.

The conclusion needs to be strengthened by restating the main findings and contributions, as well as the significance and future potential of the study in a concise yet compelling manner. For example, "This study provides evidence for reduced prefrontal activation during cognitive control in elderly with depressive symptoms. The Stroop-fNIRS paradigm shows promise as an objective indicator of depressive symptoms and may aid diagnosis, assessment and treatment monitoring in geriatric depression."

Avoid overuse of "this study" and "the present study". Use alternative expressions for variation.

Comments on the Quality of English Language

The language needs to be thoroughly proofread to check for any grammatical, spelling or punctuation errors.

The sentence construction needs to be simplified by reducing the length and complexity. Some sentences contain multiple clauses that can cause confusion or reduce clarity. It is better to break up such lengthy sentences into shorter, separate sentences.

Transitional words need to be used to create better flow and coherence between ideas, especially when moving from one topic to another. Words/phrases such as "moreover", "in addition", "on the other hand" can help to link sentences and paragraphs.

The language needs to be made more concise by removing redundancy or repetition. Some information is repeated multiple times across sections. Ensure that all details are stated only once to avoid unnecessary lengthiness.

The tone needs to be made more objective by reducing the use of personal pronouns (e.g. "I", "we"). Since this is an academic research paper, it is better to focus on discussing the study and results itself rather than personal viewpoints. Expressions like "the study found that" or "the results showed" can be used instead.

Author Response

Response to Reviewers’ Comments

July 24th, 2023

Dear Reviewer,

We greatly appreciate your thoughtful comments, which helped us improve our manuscript's quality and readability. The authors agreed with the points made and have revised the manuscript as requested.

Abstract

The aim needs to be more specifically stated, e.g. "To investigate differences in prefrontal cortex activation between elderly with and without depressive symptoms during cognitive tasks using functional near-infrared spectroscopy (fNIRS)."

Thank you for your suggestion. We revised the aim of the study as follows:

“To investigate differences in prefrontal cortex activation between elderly with and without depressive symptoms during cognitive tasks using functional near-infrared spectroscopy (fNIRS).”

The methods section should concisely state the sample size, inclusion/exclusion criteria, assessment tools/measures and experimental tasks used. For example,  "204 elderly participants without psychiatric or neurological disorders completed the Geriatric Depression Scale, digit span, Verbal Fluency Test and Stroop test while prefrontal cortex activation was recorded using fNIRS."

Thank you for your suggestion. We revised the aim of the study as follows:

“204 elderly participants without psychiatric or neurological disorders completed the Geriatric Depression Scale, digit span, Verbal Fluency Test and Stroop test while prefrontal cortex activation was recorded using fNIRS.”

The conclusion needs to be a strong, compelling summary statement regarding the key implications or significance of the results.

We revised the lines in the conclusion as follows:

In conclusion, this study provides evidence for reduced prefrontal activation during cognitive control in elderly with depressive symptoms. The Stroop-fNIRS paradigm shows promise as an objective indicator of depressive symptoms and may aid in diagnosis, assessment, and treatment monitoring in geriatric depression. Future research should control possible variables with larger sample sizes.”

Avoid excessive use of abbreviations in the abstract. Only use if unavoidable and spell out in full at first mention.

Thank you for your suggestion. We have revised the abstract to avoid the use of abbreviations.

Introduction

The first paragraph briefly introduces the background of depression in elderly and challenges in diagnosis and treatment. However, it lacks a clear thesis statement to specify the objectives and focuses of the current study. It is recommended to add a thesis statement at the end of the first paragraph to guide the readers.

Thank you for your suggestion. We have added lines to clarify this study’s objectives and focuses as follows:

The present study’s primary objective was to investigate the presence of depressive symptoms, while not necessarily enough to fulfill MDD criteria, may impact pre-frontal hemodynamic response.”

The second paragraph should be reorganized. Currently, it introduces NIRS and findings of previous studies first and then states the objectives of the current study. For clarity and logic, it is better to first state the objectives and research questions, and then explain how NIRS can be used to address the research questions with support from previous findings.

Thank you for your suggestions. We have revised expressions, edited English, and changed the sentence order to better clarify the study’s objective.

The language needs to be further polished with better coherence and flow. Some sentences are too long and difficult to follow. Transition words can be used to enhance the coherence between ideas.

For conciseness, the background information on PET and fMRI in the second paragraph could be removed. The authors could just focus on introducing NIRS and its applications in investigating MDD.

Thank you for your suggestions. We have revised the English as you commented. Also, we removed the background information on PET and fMRI in the second paragraph to focus on introducing NIRS.

Methods

The method section needs to be substantially expanded to provide sufficient details for replication. Important information such as specific inclusion/exclusion criteria, equipment specifications, and data analysis procedures are missing or not elaborated enough.

Thank you for your opinion. For inclusion/exclusion criteria, we added lines, as shown below.

“All participants were recruited amongst volunteers for blood donations and could answer a self-questionnaire and perform cognitive tasks. All participants were interviewed to exclude comorbid medical conditions, neurodegenerative diseases, and psychiatric diseases.”

We have described equipment specification and data analysis procedures in 2.3. experimental design section, in lines 90-92, as “We recorded the hemodynamic response during the performance of the protocol using a commercial wireless continuous-wave near-infrared spectroscopy system (NIRSIT; OBELAB Inc., Seoul, Republic of Korea)” (Choi, J.; Kim, J.; Hwang, G.; Yang, J.; Choi, M.; Bae, H. Time-Divided Spread-Spectrum Code-Based 400 fW-Detectable Multichannel fNIRS IC for Portable Functional Brain Imaging. IEEE Journal of Solid-State Circuits 2016, 51, 484-495, doi:10.1109/JSSC.2015.2504412.), and added lines as below for data analysis procedures.

“”

For the participants, more demographic information such as age, gender, education level, and socioeconomic status needs to be provided. The sample size for each group should be clearly stated. The inclusion/exclusion criteria should be specified in a list format.

Thank you for your comments. We added the description of demographic information on 3. Results as below

Among 206 participants, 117 were in control, and 67 were in the depressive symptom group. The average age for the control group was 72 years old, and the de-pressive-symptom group average was 71 years old, with no significant differences (p>0.05) between groups. There were no statistically significant differences in sex and education between the depressive symptom and control groups. Expectedly, the depressive symptom group had a higher score on the GDS (p<0.001).

Inclusion/exclusion criteria are listed in 2.1. participants section as below

Two hundred and six people over 60 years old were recruited in the Biobank of the Veterans Medical Research Institute. All participants were recruited amongst volunteers for blood donations and could answer a self-questionnaire and perform cognitive tasks. All participants were interviewed to exclude comorbid medical conditions, neurodegenerative diseases, and psychiatric diseases. Study participants were excluded if they reported cognitive impairment, hearing impairment, severe cardiopulmonary diseases, unstable medical illnesses, or a history of psychiatric disorders. Participants taking antidepressants, antipsychotics, and hypnotics were also excluded.

For the GDS, the validity, reliability and scoring procedures of the scale should be briefly described. The cut-off score of 5 for defining depression also needs to be justified with evidence from the literature.

We added explanation regarding GDS questionnaire as below in 2.2 Assessment of depression symptoms

The GDS is the most widely used self-rating scale for screening for depression in elderly patients. It comprises 15 dichotomous depression items with total cores ranging from 0 to 15; higher scores indicate more severe depressive symptoms1. In other studies of the GDS, sensitivity and specificity ranged from 79% to 100% and 67% to 80%, respectively, in primary care elderly2. Korean version of GDS was validated for use in Korean elderly, and a score of 8 was suggested as the optimal cut-off point to screen for MDD3

  1. Burke WJ, Roccaforte WH, Wengel SP. The short form of the Geriatric Depression Scale: a comparison with the 30-item form. J Geriatr Psychiatry Neurol 1991;4:173-178.
  2. Watson LC, Pignone MP. Screening accuracy for late-life depression in primary care: a systematic review. J Fam Pract 2003;52:956-964.
  3. Bae JN, Cho MJ. Development of the Korean version of the Geriatric Depression Scale and its short form among elderly psychiatric patients. Journal of Psychosomatic Research 2004;57:297-305.

For the experimental design, the specific cognitive tasks, duration, trial numbers and intervals need to be elaborated in detail. The reasons for choosing the specific tasks need to be explained. Diagrams or pictures can be used to illustrate the tasks design and procedures.

Thank you for your opinion. We have described the cognitive tasks used in this study in 2.3. Experimental design.

For the NIRS system, the full name of the model, spatial resolution, wavelengths used and sampling rate should be provided in the first mention. Additional technical specifications are needed for the sources, detectors and the modified Beer-Lambert law used for data analysis.

Thank you for your opinion. We have added explanation regarding, the full name of the model, spatial resolution, wavelengths used and sampling rate, sources, detectors and modified Beer-Lambert law used for data analysis in 2.4. Data processing.

For data analysis, the specific statistical tests used for which types of comparisons need to be clearly stated. The reasons for choosing non-parametric tests should be explained based on the Shapiro-Wilk test results. Multiple comparison correction methods should be considered if there are many group comparisons.

Thank you for your opinions. We underwent Shapiro-Wilk test to check normality, and performed Wilcoxon signed-rank tests which is non-parametric tests since all values in both groups were not normally distributed. Since there were two groups, control and depressive symptom group, multiple group comparison was not applied. We described details in 2.5. statistical analysis.

Results

The results need to be presented in a logical order in separate paragraphs with smooth transition. It is suggested to start with participant demographics, then cognitive performance and followed by NIRS findings. Currently, the information is mixed together, reducing readability. For demographics, report mean (standard deviation) for age, and percentage for gender and education. Provide p values to compare groups. For cognitive performance, report mean (SD, range) scores for each task. Provide test statistics and p values for between-group comparisons.

Thank you for your valuable comments. We rearranged the table and figures to improve readability and clearly stated the demographics, cognitive performance, and NIRS findings in 3. Result section. We also provided test statistics and p values of demographics, and cognitive performances in Table 3

Table 1. Demographics and Task performance of cognitive tasks

Control

Depressive symptom

P

Number of participants

117

67

Female, n (%)

82 (70.1)

54 (80.6)

0.119

Age, year, median (IQR)

72 (67-74)

71 (67-73)

0.484

Education, year, mean

GDS, mean

9.90 (3.65)

1.65 (1.26)

8.16 (4.26)

7.47 (2.67)

0.203

<0.001

Task performance

Digit span backward, median (IQR)

VFT, mean

Stroop WR, median (IQR)

Stroop CR, median (IQR)

Stroop WR-CR, median (IQR)

4 (3.5-4.0)

11.01 (3.31)

24.5 (20.0-25.0)

10.0 (7.0-19.0)

10.0 (4.0-16.0)

4 (3.0-4.0)

10.08 (2.69)

23.0 (19.0-25.0)

12.0 (8.0-18.0)

8.0 (3.0-13.0)

0.025

0.061

0.058

0.403

0.089

Data are presented as number (SD) unless otherwise specified.

CR, color reading; IQR, interquartile range; SD, standard deviation; VFT, verbal fluency task; WR, word reading may have a footer.

Check figures for titles.

Thank you for your comment. We revised the figure titles.

Discussion and conclusion

The discussion needs to have a clear and logical flow of ideas. Currently, the points are not well-connected, making the discussion appear disorganized and lacking in coherence. It is suggested to start with restating the main findings, then compare with previous literature, discuss implications, limitations, and future directions in separate paragraphs.

For the main findings, restate the differences in prefrontal activation during the Stroop test and the lack of differences during VFT and digit span between groups. Elaborate on the implications of these results in light of what is known about these tasks and prefrontal cortex function. Discuss the potential underlying neural mechanisms.

Thoroughly compare the current results with findings from previous fNIRS, fMRI and neuropsychological studies on depression, MCI and executive function. Explain how the results support or differ from the existing literature. Discuss the clinical implications and utilities of the Stroop test and fNIRS.

In limitations. Discuss how the lack of neuropsychological assessment, medical evaluation and control of confounding factors may have affected the results and interpretation. Suggest improvements for future research.

Thank you for your valuable opinions. I have thoroughly rearranged and rewrote the discussion section. I have discussed the stroop test and its mechanism, previous studies, and other neuroimaging studies in MDD patients.

The conclusion needs to be strengthened by restating the main findings and contributions, as well as the significance and future potential of the study in a concise yet compelling manner. For example, "This study provides evidence for reduced prefrontal activation during cognitive control in elderly with depressive symptoms. The Stroop-fNIRS paradigm shows promise as an objective indicator of depressive symptoms and may aid diagnosis, assessment and treatment monitoring in geriatric depression."

Thank you for your opinions. I have revised the conclusion part as follows:

“In conclusion, this study provides evidence for reduced prefrontal activation during cognitive control in elderly with depressive symptoms. The Stroop-fNIRS paradigm shows promise as an objective indicator of depressive symptoms and may aid in diagnosis, assessment, and treatment monitoring in geriatric depression. Future research should control possible variables with larger sample sizes.”

Avoid overuse of "this study" and "the present study". Use alternative expressions for variation.

Thank you for your suggestions. We have revised the English as you commented.

Comments on the Quality of English Language

The language needs to be thoroughly proofread to check for any grammatical, spelling or punctuation errors.

The sentence construction needs to be simplified by reducing the length and complexity. Some sentences contain multiple clauses that can cause confusion or reduce clarity. It is better to break up such lengthy sentences into shorter, separate sentences.

Transitional words need to be used to create better flow and coherence between ideas, especially when moving from one topic to another. Words/phrases such as "moreover", "in addition", "on the other hand" can help to link sentences and paragraphs.

The language needs to be made more concise by removing redundancy or repetition. Some information is repeated multiple times across sections. Ensure that all details are stated only once to avoid unnecessary lengthiness.

The tone needs to be made more objective by reducing the use of personal pronouns (e.g. "I", "we"). Since this is an academic research paper, it is better to focus on discussing the study and results itself rather than personal viewpoints. Expressions like "the study found that" or "the results showed" can be used instead.

Thank you for your valuable comments. We have revised the English line by line as you commented.

In summary, we revised our manuscript to address the various issues raised by the reviewers. Upon reviewing our revisions, we hope you will find the manuscript acceptable for publication in Brain Sciences.

Sincerely yours,

YoungSoon Yang, MD

May 24th, 2023

YoungSoon Yang, MD

Department of Neurology, Soonchunhyang University College of Medicine, Cheonan Hospital

31 Soonchunhyang 6-gil, Dongnam-gu, Cheonan 31151, Republic of Korea

Tel: +82-41-570-3639

Fax: +82-41-570-3639

Email: astro76@naver.com

Reviewer 4 Report

Comments and Suggestions for Authors

This study aimed to analyze and compare frontal cortex activation using functional near-infrared spectroscopy (fNIRS) in elderly participants with and without depressive symptoms. A total of 204 community-based individuals aged 60-80 years, without psychiatric disorders or dementia, were included. Geriatric Depression Scale (GDS) and fNIRS were utilized to assess depression presence, along with cognitive tasks such as the digit span, Verbal Fluency Task (VFT), and Stroop test to examine hemodynamic response in the frontal cortex. The results revealed significantly reduced hemodynamics during the Stroop test in the depressive-symptom group. The mean accΔHbO2 levels were lower in the depressive-symptom group compared to the control group, both in the overall channel averages and right hemisphere averages. No significant differences were found in hemodynamic response between the two groups during the digit span backwards and VFT tasks. These findings suggest that the hemodynamic response, particularly during the Stroop test, differs significantly between individuals with and without depressive symptoms, indicating decreased frontal cortex activation in the depressive-symptom group.

The article considers individuals with depressive symptoms without a clinical diagnosis made by a specialist, and this approach should be approached with caution. Depressed individuals were defined based on scores above 5 on the Geriatric Depression Scale (GDS), which have been shown to best predict depression and poor perceived quality of life. However, it is important to note that relying solely on self-report questionnaires and cutoff scores may not provide a definitive assessment of clinical depression. The absence of a clinical diagnosis by a specialist, such as a psychiatrist or psychologist, leaves room for potential misclassification or inclusion of individuals who may not meet the criteria for a formal diagnosis of depression. 

Caution should also be exercised when interpreting the statement suggesting impaired recruitment of the prefrontal cortex in conflict resolution in the depressive-symptom group compared to the control group. While the study observed lower activation levels in the prefrontal cortex during the color task, it is important to note that this finding alone does not provide conclusive evidence of impaired recruitment. Alternative explanations should be considered, as there could be various factors influencing the observed differences, such as variations in task performance, cognitive strategies, or individual differences in brain functioning. Further research is needed to explore and confirm the underlying mechanisms and potential alternative explanations for the observed differences in prefrontal cortex activation during conflict resolution tasks between individuals with depressive symptoms and the control group.

An alternative explanation for the observed discrepancy in results between the VFT and digit span tasks, which showed no statistical significance, and the activation map showing visible differences between the depressive-symptom and control groups, could be limitations in the sensitivity of fNIRS to detect subtle differences in frontal activation patterns. While previous studies using fNIRS have reported a reduction in frontal oxy-Hb response during the VFT in depression, and higher oxy-Hb response during the digit span task, it is possible that fNIRS may not be as effective in capturing the nuanced differences in frontal activation in individuals with subsyndromal depression.

The conclusion section is brief and could benefit from either expanding upon the conclusive thoughts or merging it with the discussion section for a more comprehensive summary. This would provide a thorough analysis of the implications and alternative explanations for the observed differences in prefrontal cortex activation. Such revisions would enhance the clarity and completeness of the research presentation.

Comments on the Quality of English Language

It seems OK

Author Response

Response to Reviewers’ Comments

July 24th, 2023

Dear Reviewer,

We greatly appreciate your thoughtful comments, which helped us improve our manuscript's quality and readability. The authors agreed with the points made and have revised the manuscript as requested.

This study aimed to analyze and compare frontal cortex activation using functional near-infrared spectroscopy (fNIRS) in elderly participants with and without depressive symptoms. A total of 204 community-based individuals aged 60-80 years, without psychiatric disorders or dementia, were included. Geriatric Depression Scale (GDS) and fNIRS were utilized to assess depression presence, along with cognitive tasks such as the digit span, Verbal Fluency Task (VFT), and Stroop test to examine hemodynamic response in the frontal cortex. The results revealed significantly reduced hemodynamics during the Stroop test in the depressive-symptom group. The mean accΔHbO2 levels were lower in the depressive-symptom group compared to the control group, both in the overall channel averages and right hemisphere averages. No significant differences were found in hemodynamic response between the two groups during the digit span backwards and VFT tasks. These findings suggest that the hemodynamic response, particularly during the Stroop test, differs significantly between individuals with and without depressive symptoms, indicating decreased frontal cortex activation in the depressive-symptom group.

The article considers individuals with depressive symptoms without a clinical diagnosis made by a specialist, and this approach should be approached with caution. Depressed individuals were defined based on scores above 5 on the Geriatric Depression Scale (GDS), which have been shown to best predict depression and poor perceived quality of life. However, it is important to note that relying solely on self-report questionnaires and cutoff scores may not provide a definitive assessment of clinical depression. The absence of a clinical diagnosis by a specialist, such as a psychiatrist or psychologist, leaves room for potential misclassification or inclusion of individuals who may not meet the criteria for a formal diagnosis of depression.

Thank you for your valuable comment. We agree that not assessing the participants’ cognitive state is a limitation in our research. We acknowledge that self-report measures alone are insufficient to rule out neurocognitive disorders definitively. In retrospect, incorporating a comprehensive cognitive assessment would have been valuable in ensuring a more accurate evaluation of cognitive functioning among the participants. In this study's clinical setting, it was impossible to provide the human and temporal environment to perform the cognitive test on the subjects directly. We emphasized this point in limitation part in discussion.

Caution should also be exercised when interpreting the statement suggesting impaired recruitment of the prefrontal cortex in conflict resolution in the depressive-symptom group compared to the control group. While the study observed lower activation levels in the prefrontal cortex during the color task, it is important to note that this finding alone does not provide conclusive evidence of impaired recruitment. Alternative explanations should be considered, as there could be various factors influencing the observed differences, such as variations in task performance, cognitive strategies, or individual differences in brain functioning. Further research is needed to explore and confirm the underlying mechanisms and potential alternative explanations for the observed differences in prefrontal cortex activation during conflict resolution tasks between individuals with depressive symptoms and the control group.

Thank you for your thoughtful review of the study regarding impaired recruitment of the prefrontal cortex in conflict resolution among individuals with depressive symptoms compared to the control group. I agree with your suggestion that further research is needed to confirm the observed differences in prefrontal cortex activation during conflict resolution tasks between the groups. By conducting more extensive studies, researchers can address the limitations of the current study and investigate potential alternative explanations. This would enhance our understanding of the complex relationship between depressive symptoms and prefrontal cortex functioning. I addressed direction for future study in the discussion part.

An alternative explanation for the observed discrepancy in results between the VFT and digit span tasks, which showed no statistical significance, and the activation map showing visible differences between the depressive-symptom and control groups, could be limitations in the sensitivity of fNIRS to detect subtle differences in frontal activation patterns. While previous studies using fNIRS have reported a reduction in frontal oxy-Hb response during the VFT in depression, and higher oxy-Hb response during the digit span task, it is possible that fNIRS may not be as effective in capturing the nuanced differences in frontal activation in individuals with subsyndromal depression.

Thank you for your thoughtful review. We acknowledge that most previous studies applied VFT for assessing fNIRS and demonstrated differences in frontal activation in depression group. We agree that VFT and digit span test may not be effective in capturing differences the subsyndromal depression, and this would be important finding of the present study. I stated this point in discussion part.

The conclusion section is brief and could benefit from either expanding upon the conclusive thoughts or merging it with the discussion section for a more comprehensive summary. This would provide a thorough analysis of the implications and alternative explanations for the observed differences in prefrontal cortex activation. Such revisions would enhance the clarity and completeness of the research presentation.

Thank you for your valuable comments. We have revised the conclusion section as you suggested.

In summary, we revised our manuscript to address the various issues raised by the reviewers. Upon reviewing our revisions, we hope you will find the manuscript acceptable for publication in Brain Sciences.

Sincerely yours,

YoungSoon Yang, MD

May 24th, 2023

YoungSoon Yang, MD

Department of Neurology, Soonchunhyang University College of Medicine, Cheonan Hospital

31 Soonchunhyang 6-gil, Dongnam-gu, Cheonan 31151, Republic of Korea

Tel: +82-41-570-3639

Fax: +82-41-570-3639

Email: astro76@naver.com

Round 2

Reviewer 1 Report

Comments and Suggestions for Authors

The review of the revised version of the manuscript entitled: “fNIRS Assessment During Cognitive Tasks in Elderly Patients with Depressive Symptoms

Comments for Authors:

Thank you for the precise revision you have done. The authors made their best effort to edit the manuscript based on comments and therefore, the paper is now much improved. However, the limitation which is about the possibility of presence of neurocognitive disorders in participants still cofounds the results and conclusions of such a valuable study.

Good luck

Reviewer 3 Report

Comments and Suggestions for Authors

None